# Human promoter directionality is determined by transcriptional initiation and the opposing activities of INTS11 and CDK9

**Joshua D Eaton[†‡](), Jessica Board[†](), Lee Davidson, Chris Estell, Steven West\***

The Living Systems Institute, University of Exeter, Exeter, United Kingdom

**Abstract** RNA polymerase II (RNAPII) transcription initiates bidirectionally at many human protein-coding genes. Sense transcription usually dominates and leads to messenger RNA production, whereas antisense transcription rapidly terminates. The basis for this directionality is not fully understood. Here, we show that sense transcriptional initiation is more efficient than in the antisense direction, which establishes initial promoter directionality. After transcription begins, the opposing functions of the endonucleolytic subunit of Integrator, INTS11, and cyclin-dependent kinase 9 (CDK9) maintain directionality. Specifically, INTS11 terminates antisense transcription, whereas sense transcription is protected from INTS11-dependent attenuation by CDK9 activity. Strikingly, INTS11 attenuates transcription in both directions upon CDK9 inhibition, and the engineered recruitment of CDK9 desensitises transcription to INTS11. Therefore, the preferential initiation of sense transcription and the opposing activities of CDK9 and INTS11 explain mammalian promoter directionality.

## eLife assessment

The **important** study uses a new experimental method to provide **compelling** evidence on how sense- and anti-sense transcription is differentially regulated. The method described here can generally be used to study the alterations in transcription. This article will be of interest to scientists working in the gene regulation community.

## Introduction

In humans, most protein-coding gene promoter regions initiate transcription bidirectionally. Usually, only sense transcription efficiently elongates and leads to messenger (m)RNA synthesis. Antisense transcription terminates within a few kilobases (kb), and the short non-coding (nc)RNA is degraded (*Preker et al., 2008*). Similar asymmetry frequently occurs in unicellular eukaryotes (e.g. budding yeast) and at some plant promoters (*Wyers et al., 2005*; *Thieffry et al., 2020*). Thus, promoter directionality is a broadly observed phenomenon. Interestingly, bidirectionality is the ground state of promoters and directionality is acquired over evolutionary time (*Jin et al., 2017*). This is proposed to be through a combination of DNA sequences and proteins that favour the directional initiation and elongation of transcription.

The present explanation for the directionality of mammalian RNAPII promoters involves the arrangement of U1 snRNA binding sites and polyadenylation signals (PASs). U1 snRNA promotes RNAPII elongation through protein-coding genes by binding to RNA and preventing early termination (*Kaida et al., 2010*; *Mimoso and Adelman, 2023*; *Chiu et al., 2018*). RNA-bound U1 prevents early termination by inhibiting PASs and antagonising other attenuation mechanisms, which include those

**\*For correspondence:**
S.West@exeter.ac.uk

[†]()These authors contributed equally to this work

**Present address:** [‡]()MRC Laboratory of Molecular Biology, Cambridge Biomedical Campus, Francis Crick Ave, Trumpington, United Kingdom

**Competing interest:** The authors declare that no competing interests exist.

involving the PP1-PNUTS phosphatase and Restrictor complexes (*So et al., 2019*; *Estell et al., 2023*). In contrast, U1 binding sites are rarer within short antisense transcripts, which are often rich in PAS sequences that are proposed to promote transcriptional termination (*Almada et al., 2013*). As such, this model predicts that polyadenylation factors control a large fraction of antisense transcriptional termination.

The multi-subunit Integrator complex also regulates promoter-proximal transcription (*Wagner et al., 2023*; *Lykke-Andersen et al., 2021*; *Beckedorff et al., 2020*; *Tatomer et al., 2019*; *Elrod et al., 2019*). The Integrator complex comprises the backbone, arm/tail, phosphatase, and endonuclease modules (*Wagner et al., 2023*). Its endoribonuclease is INTS11, and its phosphatase activity is mediated by an association between INTS6 and protein phosphatase 2A (PP2A). Loss of the INTS11 endonuclease broadly affects promoter-proximal transcriptional attenuation, whereas INTS6-PP2A activity is proposed to regulate the escape of RNAPII into elongation (*Hu et al., 2023*; *Stein et al., 2022*). INTS6-PP2A phosphatase functionally antagonises CDK9, which is vital for RNAPII promoter escape and productive elongation (*Vervoort et al., 2021*). By this model, CDK9 activity promotes elongation through protein-coding genes and INTS6-PP2A opposes it.

Here, we tested the prediction that polyadenylation factors control transcription directionality by terminating antisense transcription, but this is not usually the case. Instead, we find that promoter directionality is often conferred by preferential initiation in the sense direction and is thereafter maintained by INTS11 and CDK9. The termination of antisense transcription is constitutively INTS11-dependent, whereas most sense transcription is hypersensitive to INTS11 only when CDK9 activity is simultaneously inhibited. We hypothesise that CDK9 activity protects sense transcription from attenuation by INTS11 and that reduced CDK9 activity in the antisense direction exposes RNAPII to INTS11-dependent termination.

## Results

### Polyadenylation factor depletion does not generally increase or extend antisense transcription

The current explanation of mammalian promoter directionality invokes early PAS-dependent termination of antisense transcription (*Almada et al., 2013*). This is partly based on the direct detection of polyadenylated antisense transcripts, which provides evidence that some of their transcriptional termination is PAS-dependent. However, there are many non-polyadenylated antisense RNAs that might be attenuated in other ways (*Gockert et al., 2022*). To test the contribution of PAS-dependent termination towards antisense transcriptional termination and promoter directionality, we tagged *RBBP6* with the dTAG degron in HCT116 cells. Because RBBP6 is required to activate the PAS cleavage machinery (*Schmidt et al., 2022*; *Boreikaite et al., 2022*), its depletion should inhibit any PAS-dependent transcriptional termination. Three homozygous *dTAG-RBBP6* clones were isolated (*Figure 1—figure supplement 1A*) and tagged RBBP6 was efficiently depleted after exposure to the dTAGv-1 degrader (*Figure 1A*). To test the contribution of RBBP6 to nascent transcription, we used POINT (Polymerase Intact Nascent Transcript)-seq (*Sousa-Luís et al., 2021*), which maps full-length RNA extracted from immunoprecipitated RNAPII.

*Figure 1B* shows POINT-seq coverage over *NEDD1* including the upstream antisense transcript. As expected, RBBP6 loss causes a transcriptional termination defect beyond the *NEDD1* PAS shown by the extended POINT-seq signal downstream of the gene. However, RBBP6 does not affect the termination of upstream antisense RNA based on this not being extended when RBBP6 is depleted. Meta-analysis of this data on genes separated from their neighbours by ≥10 kb shows that RBBP6 loss causes a general increase in transcription downstream of the sense gene PAS but has little effect on upstream antisense transcription (*Figure 1C and D*). Although many antisense transcripts contain multiple AAUAAA PAS sequences that could aid termination, our POINT-seq experiment suggests that RBBP6 depletion has limited impact on the transcription of these RNAs (*Figure 1—figure supplement 1B and C*). Similarly, we recently showed that the PAS-dependent 5'→3' exonucleolytic torpedo terminator, XRN2, does not affect antisense transcriptional termination (*Estell et al., 2023*). Therefore, although some antisense transcripts are polyadenylated, most antisense transcription appears to terminate using PAS-independent mechanisms. Consequently, PAS-dependent termination cannot fully explain the promoter-proximal attenuation of antisense transcription.

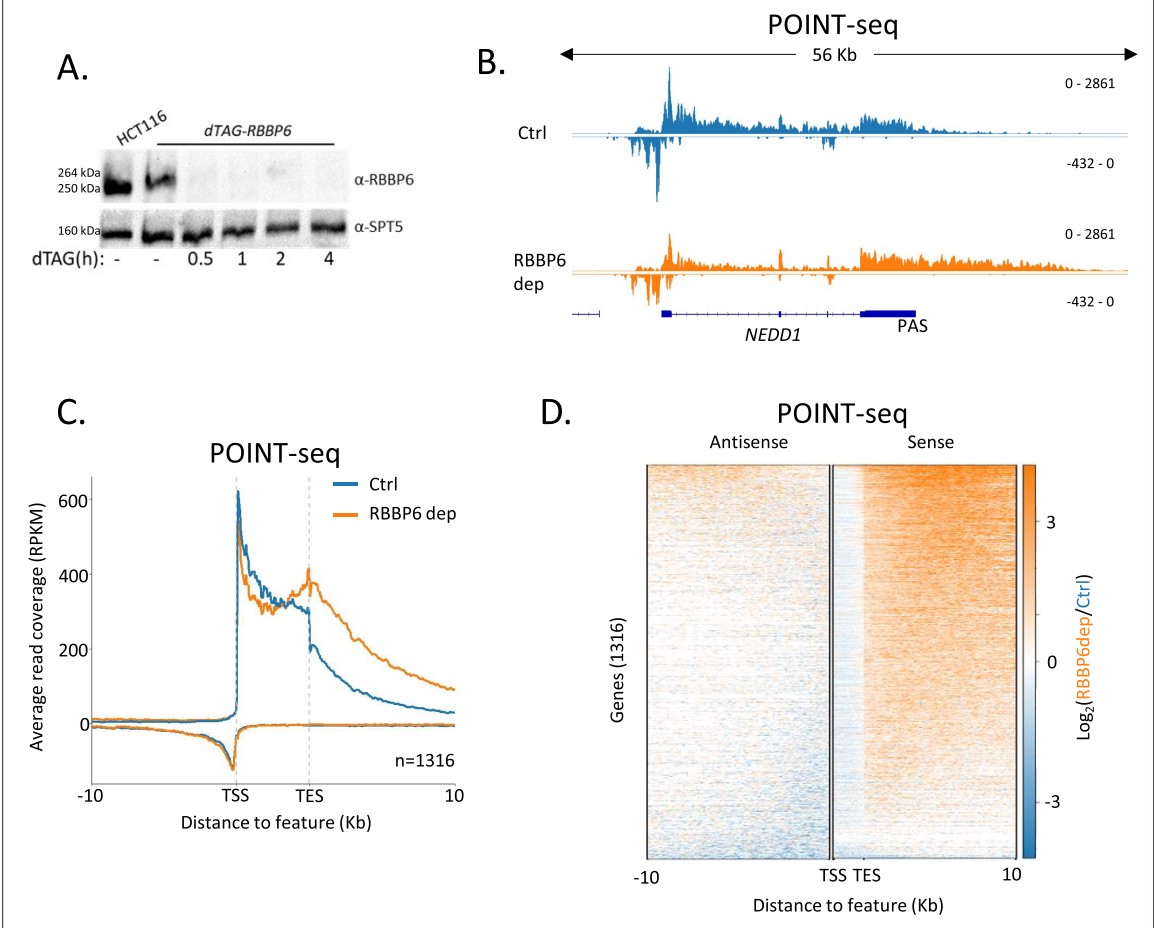

**Figure 1.** RBBP6 loss disrupts polyadenylation signal (PAS)-dependent termination of sense transcription. (**A**) Western blot demonstrating the depletion of dTAG-RBBP6 over a time course of dTAGv-1 addition. SPT5 serves as a loading control. (**B**) Genome browser track of *NEDD1* in POINT-seq data from *dTAG-RBBP6* cells treated (RBBP6 dep) or not (Ctrl) with dTAGv-1 (2 hr). RBBP6 depletion induces a transcriptional termination defect in the protein-coding direction (downstream of the indicated PAS) but not the upstream antisense direction. The y-axis shows Reads Per Kilobase per Million mapped reads (RPKM). (**C**) Metaplot of POINT-seq data from *dTAG-RBBP6* cells treated (RBBP6 dep) or not (Ctrl) (2 hr) with dTAGv-1. This shows 1316 protein-coding genes selected as separated from any expressed transcription unit by ≥10 kb. Signals above and below the x-axis are sense and antisense reads, respectively. The y-axis scale is RPKM. TSS = transcription start site; TES = transcription end site (and marks the PAS position). Coverage is shown over a region between 10 kb upstream of the TSS to 10 kb downstream of the TES. This is an average of two biological replicates. (**D**) Heatmap representation of the data in (**C**), which displays signal as a log2 fold change (log2FC) in RBBP6 depleted vs. un-depleted (Ctrl) conditions. This is an average of two biological replicates.

The online version of this article includes the following source data and figure supplement(s) for figure 1:

**Source data 1.** Original unannotated and uncropped images of the western blots used for *Figure 1A*.

**Source data 2.** Original uncropped images of the western blots used for *Figure 1A* with relevant bands labelled and highlighted.

**Figure supplement 1.** RBBP6 loss has limited effects on upstream antisense transcription.

**Figure supplement 1—source data 1.** Original unannotated and uncropped images of the western blots used for *Figure 1—figure supplement 1A*.

**Figure supplement 1—source data 2.** Original uncropped images of the western blots used for *Figure 1—figure supplement 1A* with relevant bands labelled and highlighted.

## Integrator depletion increases and extends antisense transcription

Our data argue that PAS-independent termination mechanisms control a large fraction of antisense transcription. A major PAS-independent termination pathway is controlled by the Integrator complex, which was first identified as the 3′ end processing complex for snRNAs (***Baillat et al., 2005***). Several more recent reports show that Integrator also terminates transcription from most other RNAPII promoters, including those that initiate antisense transcription, so it might affect promoter directionality (***Wagner et al., 2023***; ***Hu et al., 2023***; ***Stein et al., 2022***). To analyse this, we tagged its

endonucleolytic subunit (INTS11) with a dTAG degron (*Nabet et al., 2018*), which enables rapid deple-tion of its endonucleolytic subunit from HCT116 cells (*Figure 2A*). We then performed POINT-seq on *INTS11-dTAG* cells depleted or not of INTS11 to assay the global impact of INTS11 on RNAPII tran-scription. Integrator is established to control the transcription of snRNAs (*Baillat et al., 2005*), and this was detected in our POINT-seq, which demonstrates a clear extension of POINT-seq read density at *RNU5A-1* and *RNU5B-1* loci (*Figure 2B*).

As exemplified by *PGK1* (*Figure 2C*), INTS11 loss does not affect the termination of protein-coding transcription but causes a strong upregulation of upstream antisense transcription. Meta-analysis of the same gene set used for the RBBP6 POINT-seq in *Figure 1* shows the generality of antisense tran-scriptional attenuation via INTS11 (*Figure 2D*). A heatmap analysis of transcription 3 kb upstream and downstream of the transcription start sites (TSSs) of these genes demonstrates the dominant impact of INTS11 on antisense vs. sense transcription at most of these promoters (*Figure 2E*). Although the transcription of protein-coding genes is less affected by INTS11 loss, an increase in promoter-proximal transcription is apparent in some cases (*Figure 2—figure supplement 1A*). The most strongly affected examples of this are lowly expressed genes, which is consistent with recent findings (*Figure 2—figure supplement 1B*; *Lykke-Andersen et al., 2021*; *Beckedorff et al., 2020*; *Tatomer et al., 2019*; *Elrod et al., 2019*; *Hu et al., 2023*; *Stein et al., 2022*). Overall, INTS11 frequently attenuates antisense transcription, whereas a smaller fraction of sense transcription is affected.

The hypersensitivity of antisense transcripts to INTS11 might be caused by the composition of RNAPII that transcribes these regions and/or some other promoter feature. To interrogate this further, we analysed two additional promoter region classes: those where protein-coding transcripts are initi-ated in both directions and those that initiate the bidirectional transcription of unstable enhancer (e) RNAs. In the former case, transcription will be bi-directionally elongated with both transcripts resem-bling those synthesised in the sense direction at directional promoters. In the latter case, transcription in both directions is rapidly attenuated as is the case for upstream antisense transcription at directional promoters. When both directions are protein-coding, INTS11 depletion causes modest reductions in the metaplot POINT-seq signal as was the case for sense transcription in *Figure 2D* (*Figure 2—figure supplement 1C*). Conversely, eRNA transcription is upregulated upon INTS11 elimination as was the case for antisense transcription in *Figure 2* (*Figure 2—figure supplement 1D*). We conclude that short ncRNAs are more strongly affected by INTS11 than protein-coding transcripts. At directional promoters, this results in the attenuation of antisense transcription.

## Transcription initiates more efficiently in the sense direction

Although the increase of antisense POINT-seq signal following INTS11 loss is consistent with defects in transcriptional termination, it could result from increased transcriptional initiation. We were also interested in whether preferential sense initiation could contribute to mammalian promoter direction-ality as was proposed in budding yeast (*Jin et al., 2017*). To precisely resolve directional aspects of initiation and assay any impact of INTS11, we devised a variant of POINT-seq called short (s)POINT. Briefly, sPOINT follows the POINT-seq protocol, but library preparation employs the selective ampli-fication of 5′ capped RNAs <150 nts (*Figure 3A*, *Figure 3—figure supplement 1A*, 'Materials and methods'). The sPOINT signal over *PGK1* exemplifies this and demonstrates a very restricted signal close to the TSS (*Figure 3B*). *Figure 3C* compares the meta profile of POINT- and sPOINT-seq on 684 well-expressed and well-spaced (≥10 kb from neighbours) genes and highlights the full read coverage obtained by POINT-seq compared to the tightly restricted, TSS-proximal, sPOINT signal. Since most capped RNAPII-associated RNA < 150 nts maps to promoter-proximal regions, we suggest that sPOINT effectively captures and assays the well-characterised promoter-proximal RNAPII pause.

The sPOINT-seq metaplot in *Figure 3C* shows a higher signal in the sense direction in unperturbed cells, suggesting that more efficient transcriptional initiation helps to explain promoter directionality. To assay this further and test the impact of INTS11, we performed sPOINT-seq in *INTS11-dTAG* cells treated or not with dTAGv-1 and plotted sequence coverage over the promoters of well-expressed protein-coding genes (3060 promoters, *Figure 3D*). As sPOINT-seq maps the 5′ and 3′ ends of these reads, we could plot full read coverage (top plot) as well as the precise TSS position (lower plot). This analysis once again shows a higher sPOINT signal in the sense direction. In addition, the lower TSS mapped plot indicates that sense transcription initiates in a more focused manner compared to in the antisense direction. This is because a sharp sense TSS peak is evident in the lower plot of *Figure 3D*,

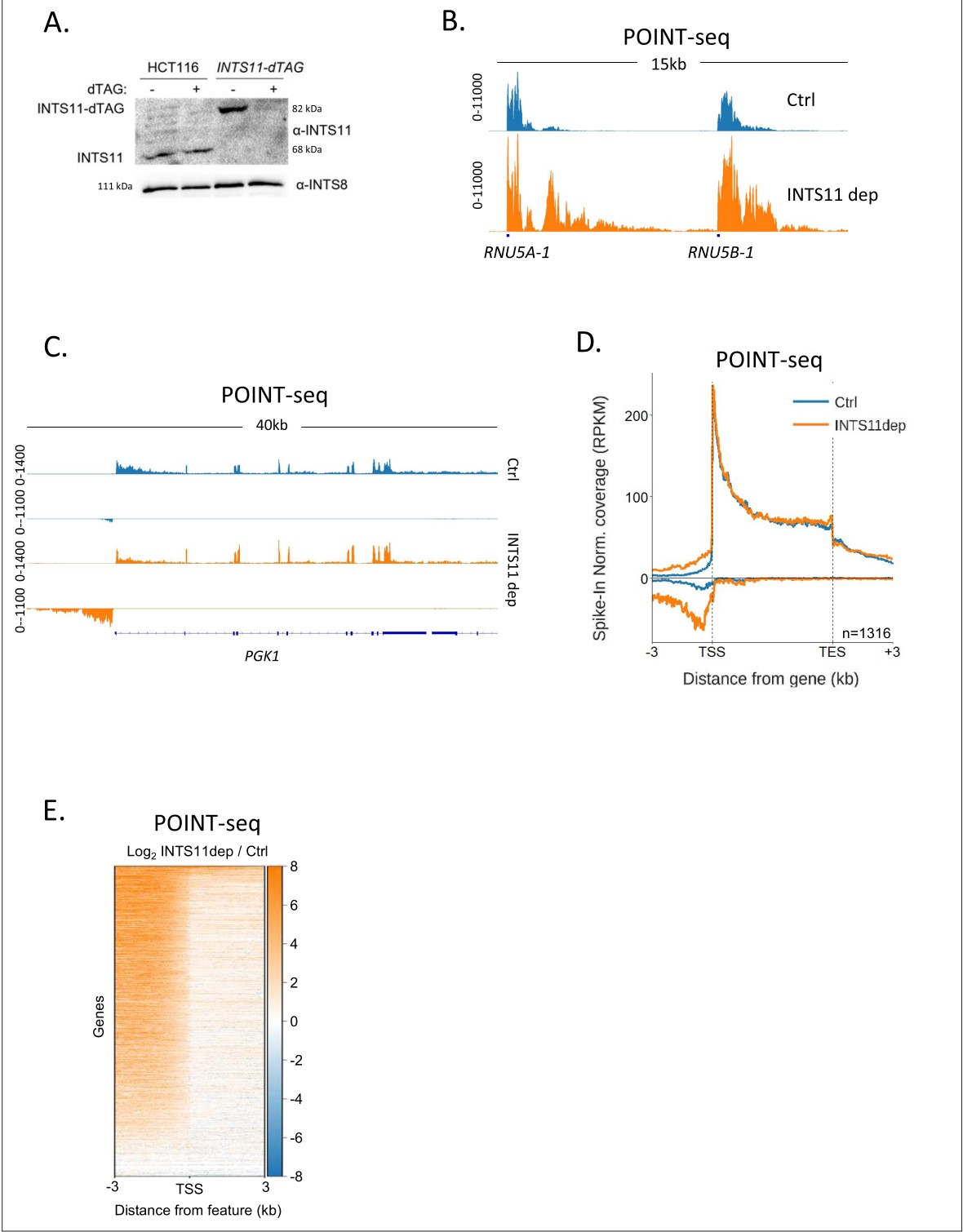

**Figure 2.** INTS11 loss disrupts the termination of antisense transcription. (**A**) Western blot demonstrating homozygous tagging of *INTS11* with dTAG and the depletion of INTS11-dTAG after 1.5 hr treatment with dTAGv-1. INTS8 serves as a loading control. (**B**) Genome browser track showing POINT-seq signal over *RNU5A-1* and *RNU5B-1* in *INTS11-dTAG* cells treated (INTS11 dep) or not (Ctrl) with dTAGv-1 (1.5 hr). Note that INTS11 depletion increases the nascent transcription signal at both loci. Y-axis shows RPKM following spike-in normalisation. (**C**) Genome browser track showing POINT-seq signal over *PGK1* and its upstream antisense region in *INTS11-dTAG* cells treated (INTS11 dep) or not (Ctrl) with dTAGv-1 (1.5 hr). Y-axis shows RPKM following spike-in normalisation. (**D**) Metaplot of POINT-seq data from *INTS11-dTAG* cells treated (INTS11 dep) or not (Ctrl) with dTAGv-1 (1.5 hr). This shows the same 1316 genes used in *Figure 1C*. Signals above and below the x-axis are sense and antisense reads, respectively. Y-axis shows RPKM

*Figure 2 continued on next page*

*Figure 2 continued*

following spike-in normalisation. Coverage is shown over a region between 3 kb upstream of the transcription start site (TSS) and 3 kb downstream of the TES. This is an average of three biological replicates. (**E**) Heatmap representation of the data in (**D**), which displays signal as a log2 fold change (log2FC) in INTS11 depleted vs. undepleted (Ctrl) conditions over a region 3 kb upstream and downstream of annotated TSSs. These POINT-seq experiments are an average of three biological replicates.

The online version of this article includes the following source data and figure supplement(s) for figure 2:

**Source data 1.** Original unannotated and uncropped images of the western blots used for *Figure 2A*.

**Source data 2.** Original uncropped images of the western blots used for *Figure 2A* with relevant bands labelled and highlighted.

**Figure supplement 1.** INTS11 loss increases the transcription of lowly expressed RNAs and eRNAs.

whereas antisense TSSs are more dispersed. Although this could be due to poor annotation of antisense vs sense TSSs, *Figure 3E* (*PGK1*) and *Figure 3—figure supplement 1B* (*ACTB*) exemplify these features on individual genes, and the metaplot in *Figure 3F* highlights the dispersed nature of antisense TSSs. Quantitation of the TSS-derived sense vs. antisense signal confirms a higher read count in the sense direction in untreated cells (*Figure 3—figure supplement 1C*). INTS11 loss caused a mild reduction in sPOINT-seq signal, suggesting that the enhanced antisense transcription seen after its elimination is not because transcriptional initiation is increased. This slight reduction in sPOINT-seq signal could result from less transcriptional initiation or be because INTS11 loss allows more RNAPII to escape the promoter and elongate transcripts beyond the ~150 nt length detectable by sPOINT-seq. Although a lower resolution technique, RNAPII ChIP-seq confirmed these promoter characteristics and the mild impact of INTS11 (*Figure 3—figure supplement 1D and E*). Overall, these data show that directionality might be initially established by more efficient initiation of sense transcription.

## CDK9 inhibition sensitises sense transcription to INTS11

After transcription initiates, additional control of directionality is evident because sense transcription still goes further than antisense transcription. Therefore, we hypothesised that sense transcription is enabled by an activity that opposes the INTS11-dependent attenuation commonly seen in the upstream antisense direction. INTS11 affects transcription very early, occupies promoters, and becomes less active as RNAPII moves into elongation (*Hu et al., 2023*; *Stein et al., 2022*; *Vervoort et al., 2021*; *Zheng et al., 2020*; *Ramamurthy et al., 1996*). Therefore, if INTS11 is counteracted to allow sense transcription, any responsible mechanism needs to act early. One of the first transcriptional checkpoints involves the phosphorylation of RNAPII and other factors by CDK9, which releases promoter-proximally paused RNAPII into elongation (*Fujinaga et al., 2023*). During this process, the Integrator-associated phosphatase, PP2A, antagonises CDK9 and presumably regulates the sensitivity of RNAPII to INTS11 (*Vervoort et al., 2021*). Because Integrator phosphatase has little effect on antisense transcription (*Hu et al., 2023*), we hypothesised that INTS11 sensitivity and CDK9 activity are inversely correlated to maintain directionality after initiation.

Our hypothesis predicts that sense transcription will be attenuated via INTS11 when CDK9 is inactive. To test this genome-wide, we depleted INTS11 from *INTS11-dTAG* cells in the presence or absence of the specific CDK9 inhibitor NVP-2 (referred to from now on as CDK9i; *Olson et al., 2018*) and performed POINT-seq. As exemplified by *TARDBP*, the depletion of INTS11 alone caused an antisense transcriptional termination defect with a milder impact on the protein-coding sense direction (*Figure 4A*). As expected, CDK9i treatment reduced transcription over the protein-coding gene body. Antisense transcription also displays CDK9 sensitivity. Importantly, in CDK9i-treated cells, INTS11 loss increased transcription in both directions. This contrasts with the dominant antisense effect deriving from just depleting INTS11 (refer to *Figure 2*). As such, CDK9 activity prevents sense transcription from being attenuated by INTS11. This is a genome-wide trend as shown in the metaplots in *Figure 4B and C*. A heatmap of the effect of INTS11 loss after CDK9 inhibition shows the bidirectional upregulation of transcription at the same promoters assayed in *Figure 2* (*Figure 4D*). Although this experiment employed longer INTS11 depletion than that in *Figure 2* (2.5 hr vs. 1.5 hr to allow concurrent CDK9 inhibition), the antisense effects remain general, and the most affected protein-coding genes strongly overlapped (*Figure 4—figure supplement 1A–C*). Importantly, when CDK9 and INTS11 are both compromised, the POINT-seq signal remains higher in the sense vs. antisense direction (black line,

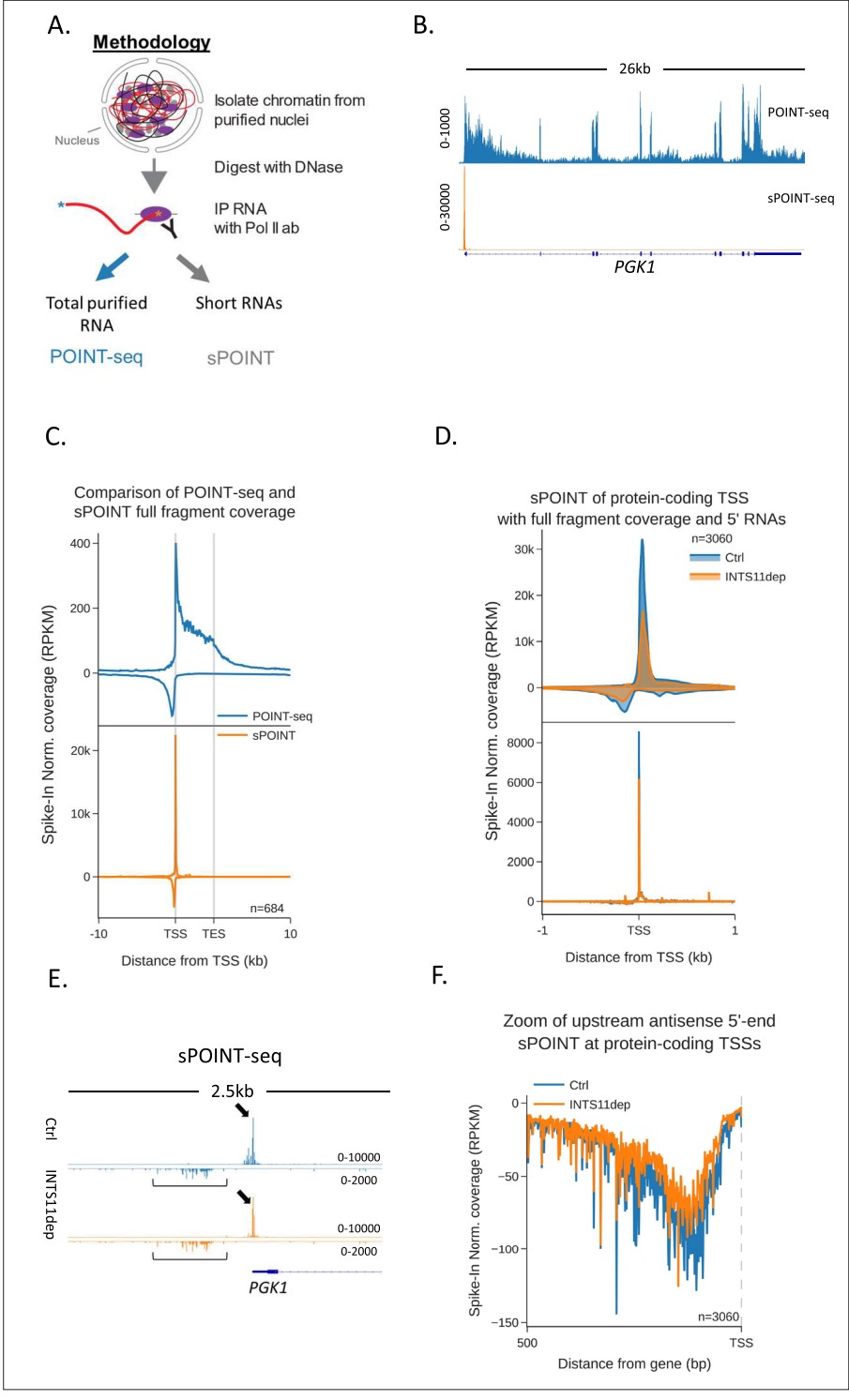

**Figure 3.** Transcription initiation is more efficient and focused in the sense direction compared to the antisense direction. (**A**) Schematic of sPOINT-seq protocol. The POINT-seq protocol is followed, in which chromatin is isolated and engaged RNAPII is immunoprecipitated. Short transcripts are preferentially amplified during library preparation (see 'Materials and methods'). (**B**) Genome browser view of POINT-seq (top trace) and sPOINT-seq (lower trace) coverage on *PGK1*. Y-axis units are RPKM. (**C**) Metaplot comparison of POINT-seq (top plot) and sPOINT-seq (lower plot) profiles across the 684 highest expressed protein-coding that are separated from expressed transcription units by ≥10 kb (top ~50%). Signals above and below the x-axis are sense and antisense reads, respectively. Coverage is shown over a region between 10 kb upstream of the transcription start site (TSS)

*Figure 3 continued on next page*

*Figure 3 continued*

and 10 kb downstream of the TES. Y-axis shows RPKM following spike-in normalisation. (**D**) Top metaplot shows full read coverage for sPOINT-seq performed in *INTS11-dTAG* cells treated (INTS11 dep) or not (Ctrl) with dTAGv-1 (1.5 hr) at the promoters of the top expressed 20% of protein-coding genes. The lower metaplot is the same data but only the 5′ end of each read is plotted. The y-axis signals are RPKM following spike-in normalisation. Coverage is shown over a region 1 kb upstream and downstream of the TSS. Two biological replicates of sPOINT were performed. (**E**) Genome browser track of *PGK1* promoter region in sPOINT-seq performed in *INTS11-dTAG* cells treated (INTS11 dep) or not (Ctrl) with dTAGv-1 (1.5 hr). This showcases the focused sense TSS (black arrows) and the dispersed antisense reads (black brackets). Note the higher y-axis scale (RPKM) for sense vs. antisense. (**F**) Metaplot zoom of the antisense TSS signals deriving from the lower plot in (**D**). This makes clear the dispersed sites of initiation. The y-axis scale is RPKM following spike-in normalisation. Coverage is shown over 500 bp upstream and antisense of the annotated sense TSS. Two biological replicates of sPOINT were performed.

The online version of this article includes the following figure supplement(s) for figure 3:

**Figure supplement 1.** Sense transcription is more efficient and focused than antisense transcription.

*Figure 4C*). This is consistent with our sPOINT-based finding that sense transcription initiates more efficiently than antisense transcription.

Following CDK9 inhibition and INTS11 loss, the largest recovery of protein-coding transcription is often over the 5′ end of genes. This presumably reflects poor elongation without CDK9 activity regardless of whether INTS11 is absent. RNAPII elongation is associated with phosphorylation of the C-terminal domain (CTD) of its largest subunit, RPB1, and most frequently occurs on Serine 5 or 2 (Ser5/2p) of its heptad repeat (*Schüller et al., 2016*; *Suh et al., 2016*). We assayed the effects of CDK9 inhibition and INTS11 depletion on these two modifications by western blotting (*Figure 4E*). As previously shown (*Olson et al., 2018*), CDK9i treatment suppresses Ser2p. However, both Ser5p and Ser2p are enhanced by INTS11 loss whether CDK9 is active or not. Thus, unattenuated transcription resulting from INTS11 depletion is associated with some CTD phosphorylation. Where CDK9i is employed, kinases that are functionally redundant with CDK9 presumably phosphorylate RNAPII.

We confirmed the CDK9-mediated suppression of INTS11 on four selected protein-coding genes using an alternative CDK9 inhibitor, 5, 6-dichloro-1-β-D-ribofuranosylbenzimidazole (DRB) (*Figure 4—figure supplement 1D*). Interestingly, inhibition of CDK7 with THZ2 also sensitised the same selected protein-coding transcripts to INTS11 attenuation (*Figure 4—figure supplement 1E*). This may be because CDK7 is required to activate CDK9 (*Larochelle et al., 2012*) or because Integrator additionally targets RNAPII when CDK7 is inactive. These data reinforce the idea that the successful execution of early transcriptional checkpoints may overcome Integrator-mediated attenuation.

If CDK9 counteracts INTS11, its recruitment should prevent transcriptional attenuation by Integrator. To assay this, we employed a plasmid where transcription is driven by the human immunodeficiency virus (HIV) promoter. Transcription from the HIV promoter results in synthesis of the trans-activating response (TAR) element and promoter proximal RNAPII pausing. Pause release requires the trans-activator of transcription (TAT), which promotes RNAPII elongation by recruiting CDK9 via TAR (*Wei et al., 1998*). INTS11 suppresses transcription from the HIV promoter when TAT is absent (*Stadelmayer et al., 2014*). To assay whether CDK9 affects this process, *INTS11-dTAG* cells were transfected with the HIV reporter with or without TAT before treatment or not with dTAGv-1. Transcription from the reporter was then analysed by qRT-PCR (*Figure 4F*). As expected, INTS11 loss induces HIV transcription in the absence of TAT. However, TAT strongly stimulates transcription (~200-fold) and desensitises it to INTS11 loss. Therefore, TAT-mediated CDK9 recruitment alleviates INTS11-dependent attenuation of transcription. Similar to endogenous protein-coding genes, CDK9 inhibition sensitises TAT-activated transcription to INTS11 (*Figure 4—figure supplement 1F*). Overall, our data strongly suggest that CDK9 activity counteracts transcriptional attenuation by INTS11.

## Discussion

Most studies on mammalian promoter directionality have focused on understanding how antisense transcription terminates and how RNAPII elongation is enabled in the sense direction. However, we show that preferential sense transcriptional initiation significantly contributes towards the directionality of many mammalian promoters. More analyses are required to elucidate the mechanism, but

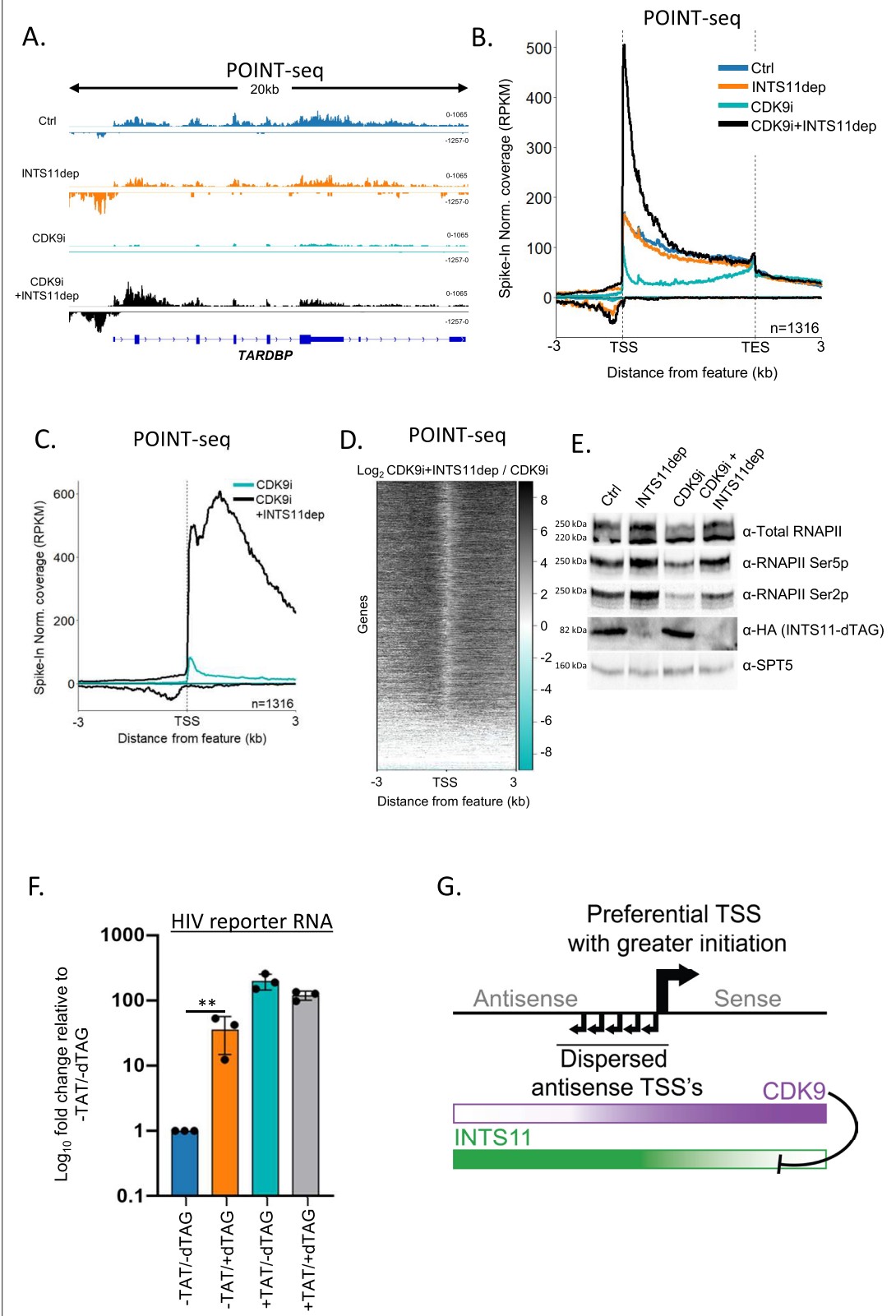

**Figure 4.** Sense transcription becomes INTS11-sensitive when CDK9 is inhibited. (**A**) Genome browser track of *TARDBP* in POINT-seq data derived from *INTS11-dTAG* cells either untreated (Ctrl), dTAG-treated (INTS11 dep), NVP-2-treated (CDK9i), or dTAG and NVP-2-treated (CDK9i+INTS11dep). Treatments were for 2.5 hr. Signals above and below the x-axis are sense and antisense reads, respectively. The y-axis scale shows RPKM following spike-in normalisation. (**B**) Metaplot of POINT-seq data derived from *INTS11-dTAG* cells either untreated (Ctrl), dTAG-treated (INTS11 dep), NVP-2-treated

*Figure 4 continued on next page*

*Figure 4 continued*

(CDK9i), or dTAG and NVP-2-treated (CDK9i+INTS11dep). Treatments were for 2.5 hr. This uses the same gene set as *Figure 1C*. Coverage is shown between 3 kb upstream of the transcription start site (TSS) and 3 kb downstream of the TES. y-axis units are RPKM following spike-in normalisation. This is an average of two biological replicates.(C) Metaplot of CDK9i and CDK9i+INTS11dep POINT-seq data but zoomed into the region 3 kb upstream and downstream of the TSS. This is an average of three biological replicates.(D) Heatmap representation of the data in (C), which displays signal as a log2 fold change (log2FC) in INTS11 depleted vs. un-depleted (Ctrl) conditions covering a region 3 kb upstream and downstream of the TSS. This is an average of three biological replicates. (E) Western blot for total RNAPII and RNAPII phosphorylated on Ser2/5 (Ser2/5p) in *INTS11-dTAG* cells treated or not with dTAGv-1 and/or NVP-2 (CDK9i). dTAG-mediated INTS11 depletion is shown in the anti-HA blot and SPT5 us used as a loading control. All treatments were for 2.5 hr. (F) qRT-PCR analysis of *INTS11-dTAG* cells transfected with the HIV reporter construct with or without TAT (±TAT) then depleted or not of INTS11 (±dTAG). Quantitation shows signals relative to those obtained in the presence of INTS11 and the absence of TAT after normalising to MALAT1 RNA levels. n = 3. Error bars show standard deviation. **p=0.01. Note that, in this experiment, INTS11 depletion was performed concurrently with transection (14 hr in total). (G) Model for promoter directionality depicting higher levels of focused transcriptional initiation in the sense direction together with opposing gradients of CDK9 and INTS11 activity that peak in sense and antisense directions, respectively.

The online version of this article includes the following source data and figure supplement(s) for figure 4:

**Source data 1.** Original unannotated and uncropped images of the western blots used for *Figure 4E*.

**Source data 2.** Original uncropped images of the western blots used for *Figure 4E* with relevant bands labelled and highlighted.

**Figure supplement 1.** CDK activity influences the sensitivity of transcription to INTS11 loss.

**Figure supplement 1—source data 1.** Original unannotated and uncropped images of the western blots used for *Figure 4—figure supplement 1G*.

**Figure supplement 1—source data 2.** Original uncropped images of the western blots used for *Figure 4—figure supplement 1G* with relevant bands labelled and highlighted.

we hypothesise that a unidirectional arrangement of at least some core promoter elements favours transcription in the protein-coding direction. Consistently, the evolution of promoter elements helps explain how a bidirectional promoter 'ground state' acquired directionality in yeast (*Jin et al., 2017*). More focused transcription initiation is also associated with highly expressed mammalian genes and with promoters that have clearly defined elements (e.g. TATA) (*Rengachari et al., 2022*). Thus, the lower level and dispersed nature of antisense initiation may reflect the suboptimal orientation of promoter elements or the opportunistic transcription of open chromatin in promoter regions.

We hypothesise that INTS11 controls transcriptional attenuation via its endonuclease activity. An inactive mutant of INTS11 (E203Q) is widely used to interrogate its catalytic function (*Baillat et al., 2005*); however, we found that it poorly associates with other Integrator components compared to wild-type INTS11 (*Figure 4—figure supplement 1G*). When employed in cells, this mutant may not always isolate the effects of INTS11 activity from those requiring an intact Integrator complex. Similarly, catalytic mutations in the highly related PAS endonuclease, CPSF73, disrupt its association with other cleavage and polyadenylation components (*Kolev et al., 2008*). Therefore, although we do not favour this hypothesis, it is formally possible that non-catalytic consequences of INTS11 loss explain the upregulation of antisense transcription.

How CDK9 activity opposes INTS11 is unresolved, but INTS11 and SPT5 are adjacent in the RNAPII: Integrator structure (*Fianu et al., 2021*). As SPT5 is a prominent substrate of CDK9 (*Yamada et al., 2006*; *Cortazar et al., 2019*; *Parua et al., 2018*), its phosphorylation might evict INTS11 or prevent its association with the complex. Consistently, Integrator is enriched on promoters, whereas phosphorylated SPT5 (SPT5p) is most prevalent during elongation over protein-coding gene bodies (*Zheng et al., 2020*; *Cortazar et al., 2019*). Furthermore, although antisense transcription is increased by INTS11 loss, published data suggest that this is not accompanied by increased Ser5p and Ser2p (*Hu et al., 2023*). As these RNAPII modifications require CDK7 and CDK9, this provides further support that these activities are less prevalent for antisense vs. sense transcription.

While the effects of INTS11 and CDK9 activities are the clearest on antisense and sense transcription, respectively, this is not binary. Some antisense transcription is lost upon CDK9 inhibition, which indicates CDK9 activity on some antisense RNAPII (*Figure 4* and *Fong et al., 2022*). Such CDK9-sensitive antisense transcription could undergo INTS11-independent attenuation, which might account for the polyadenylated antisense RNA observed previously (*Almada et al., 2013*; *Flynn et al., 2011*). Consistently, some antisense transcription is targeted by the poly(A) exosome targeting connection, which operates on polyadenylated transcripts (*Wu et al., 2020*; *Meola et al., 2016*). We and others recently described the Restrictor complex as restraining antisense transcription (*Estell et al., 2023*; *Estell et al., 2021*; *Austenaa et al., 2021*). Unlike Integrator, which associates with

NELF-bound RNAPII (*Fianu et al., 2021*), Restrictor is proximal to the distinct PAF-bound RNAPII (*Estell et al., 2023*). Thus, various complexes may target different forms of RNAPII before and after CDK9 activity. Even so, our data suggests that Integrator influences a greater volume of antisense RNAPII than the cleavage and polyadenylation pathway.

In addition to CDK9, multiple studies demonstrate a role for U1 snRNA in promoting elongation through protein-coding genes (*Kaida et al., 2010*; *Mimoso and Adelman, 2023*; *So et al., 2019*; *Vlaming et al., 2022*). U1 promotes RNAPII elongation and shields transcription from early termination via the PAS-dependent mechanism and the Restrictor complex (*Kaida et al., 2010*; *Estell et al., 2023*). The U1-mediated suppression of premature PAS-dependent termination inspired the original model of promoter directionality. As U1 sites are rarer in antisense RNAs, their termination was hypothesised to be driven by early PASs that would consequently be active. Our RBBP6 data (and other published data on PAS-associated termination factors; *Estell et al., 2023*) argues that a large fraction of antisense transcriptional termination is PAS-independent. Nevertheless, the lack of U1 sites may limit RNAPII elongation and render it sensitive to Integrator (or Restrictor). Indeed, while this paper was under consideration, U1-mediated antagonism of INTS11 was suggested to contribute to promoter directionality (*Yang et al., 2024*).

In conclusion, we provide a new model for mammalian promoter directionality and the subsequent control of bidirectional transcription (*Figure 4G*). sPOINT demonstrates more efficient sense transcription initiation, which establishes directionality. Thereafter, early events dictate the decision to elongate or attenuate transcription. In the sense direction, CDK9 activity prevents attenuation by Integrator to favour productive elongation, whereas antisense transcription is hypersensitive to INTS11 and terminates early. As INTS11 occupies most promoters and CDK9 activity is near-universally required to achieve protein-coding transcription, this model can generally explain how directionality is initiated and maintained. Our new sPOINT-seq approach will be valuable in elucidating additional aspects of promoter-proximal transcriptional regulation.

# Materials and methods

## Key resources table

| Reagent type (species) or resource | Designation | Source or reference | Identifiers | Additional information |
|---|---|---|---|---|
| Cell line (*Drosophila melanogaster*) | S2 | This paper | FLYB:FBtc0000181; RRID:CVCL_Z992 | Cell line maintained in N. Perrimon lab; FlyBase symbol: S2-DRSC |
| Cell line (*D. melanogaster*) | S2 | This paper | FLYB:FBtc0000181; RRID:CVCL_Z992 | Cell line maintained in N. Perrimon lab; FlyBase symbol: S2-DRSC |
| Cell line (*Homo sapiens*) | HCT116 | In-house | This paper | *Figures 1* and *2* |
| Cell line (*H. sapiens*) | INTS11-dTAG | In-house | This paper | *Figures 2–4* and associated figure supplements |
| Cell line (*H. sapiens*) | dTAG-RBBP6 | In-house | This paper | *Figure 1* and *Figure 1—figure supplement 1* |
| Antibody | INTS11 (rabbit polyclonal) | Abbexa | abx234340 | WB: 1:500 |
| Antibody | INTS10 (rabbit polyclonal) | Proteintech | Cat# 15271-1-AP; RRID:AB_2127260 | WB: 1:3000 |
| Antibody | INTS8 (rabbit polyclonal) | Proteintech | Cat# 18802-1-AP ; RRID:AB_10597250 | WB: 1:1000 |
| Antibody | INTS9 (rabbit polyclonal) | Proteintech | Cat# 11657-1-AP; RRID:AB_2127514 | WB: 1:1000 |
| Antibody | INTS1 (rabbit polyclonal) | Bethyl | Cat# A300-361A-T; RRID:AB_2632121 (now discontinued) | WB: 1:500 |
| Antibody | HA (rabbit monoclonal) | Cell Signalling | Cat# 3724; RRID:AB_1549585 | WB: 1:2000 |
| Antibody | RNAPII Ser2p (rat monoclonal) | Active Motif | Cat# 61084; RRID:AB_2687450 | WB: 1:2000 |
| Antibody | RNAPII Ser5p (rat monoclonal) | Millipore | Cat# 04-1572-I; RRID:AB_2801296 | WB: 1:2000 |
| Antibody | Total RNAPII (mouse monoclonal) | Hiroshi Kimura Lab | N/A | WB: 1:10,000 IP: 5–10 ug (Ab:Beads) |
| Antibody | SPT5 (mouse monoclonal) | Santa Cruz Biotech | Cat# sc-133217; RRID:AB_2196394 | WB: 1:1000 |
| Antibody | Flag (rabbit polyclonal) | Proteintech | Cat# 20543-1-AP; RRID:AB_11232216 | WB: 1:5000 |
| Antibody | Anti-rabbit (goat polyclonal) | Proteintech | Cat# SA00001-2; RRID:AB_2722564 | WB: 1:2000 |
| Antibody | Anti-mouse (goat polyclonal) | Proteintech | Cat# SA00001-1-A; RRID:AB_2890995 | WB: 1:2000 |

*Continued on next page*

*Continued*

| Reagent type (species) or resource | Designation | Source or reference | Identifiers | Additional information |
|---|---|---|---|---|
| Antibody | Anti-rat (goat polyclonal) | Thermo Fisher | Cat #A18865; RRID:AB_2535642 | WB: 1:2000 |
| Recombinant DNA reagent | px300 | Addgene | RRID: Addgene_42230 | |
| Recombinant DNA reagent | pCRIS-PITChv2-BSD-dTAG (BRD4) | Addgene | RRID:Addgene_91792 | |
| Recombinant DNA reagent | pCRIS-PITChv2-PURO-dTAG (BRD4) | Addgene | RRID:Addgene_91793 | |
| Transfected construct (HIV virus) | TAT | *Adams et al., 1988* | | |
| Transfected construct (*H. sapiens*) | INTS11 WT and E203Q | Twist | This paper | *Supplementary file 1* |
| Transfected construct (*H. sapiens*) | HIV reporter | In-house | This paper | *Figure 4* and *Figure 4—figure supplement 1* |
| Commercial assay or kit | Terminator 5-phosphate-dependent exonuclease | Lucigen | Cat# TER51020 | |
| Commercial assay or kit | SMARTer smRNA-seq kit for illumina | Takara | Cat# 635031 | |
| Commercial assay or kit | Dynabeads M280 sheep anti-mouse | Thermo Fisher | Cat# 11202D | |
| Commercial assay or kit | Anti-FLAG/DYKDDDDK magnetic beads | MilliporeSigma | Cat# M8823 | |
| Commercial assay or kit | SimpleChIP Plus Enzymatic Chromatin kit | Cell Signalling | Cat# 9005 | |
| Commercial assay or kit | NEBNext Ultra II DNA Library Prep Kit for Illumina | NEB | Cat# E7645S | |
| Commercial assay or kit | JetPRIME | PolyPlus | Cat# 114-01 | |
| Commercial assay or kit | NEBNext Ultra II Directional RNA Library Prep Kit for Illumina | NEB | Cat# E7765 | |
| Commercial assay or kit | Protoscript II reverse transcriptase | NEB | Cat# M0368 | |
| Commercial assay or kit | LUNA qPCR reagent | NEB | Cat# M3003 | |
| Commercial assay or kit | Ampure XP beads | Beckman Coulter | Cat# A63880 | |
| Other | Benzonase | Sigma | Cat# E1014-5KU | Used to digest cell extracts prior to co-immunoprecipitation (*Figure 4—figure supplement 1G*) |
| Chemical compound, drug | dTAGv-1 | Tocris | Cat# 2624313-15-9 | |
| Chemical compound, drug | NVP2 | Merck | Cat# SML3069-5MG | |
| Chemical compound, drug | THZ2 | Medchem Express | Cat# HY-12280 | |
| Chemical compound, drug | DRB | Merck | Cat# D1916 | |
| Chemical compound, drug | Trizol Reagent | Thermo Fisher | Cat# 15596026 | |
| Commercial assay or kit | Turbo DNAse kit | Thermo Fisher | Cat# AM2238 | |
| Software, algorithm | Bamtools | *Barnett et al., 2011* | NA | NA |
| Software, algorithm | BEDtools | *Quinlan and Hall, 2010* | NA | NA |
| Software, algorithm | Cutadapt | https://doi.org/10.14806/ej.17.1.200 | NA | NA |
| Software, algorithm | Deeptools | *Ramírez et al., 2016* | NA | NA |
| Software, algorithm | FastQC | https://www.bioinformatics.babraham.ac.uk/projects/fastqc/ | NA | NA |
| Software, algorithm | Hisat2 | *Kim et al., 2015* | NA | NA |
| Software, algorithm | IGV | *Robinson et al., 2011* | NA | NA |
| Software, algorithm | MACS2 | *Zhang et al., 2008* | NA | NA |
| Software, algorithm | MultiQC | *Ewels et al., 2016* | NA | NA |
| Software, algorithm | R | https://cran.r-project.org/ | NA | NA |

*Continued on next page*

*Continued*

| Reagent type (species) or resource | Designation | Source or reference | Identifiers | Additional information |
|---|---|---|---|---|
| Software, algorithm | Rstudio | https://www.rstudio.com/ | NA | NA |
| Software, algorithm | SAMTools | *Li et al., 2009* | NA | NA |
| Software, algorithm | Trim_galore! | https://github.com/FelixKrueger/TrimGalore/ (; *Krueger et al., 2023*;(RRID:SCR_011847) | NA | NA |

## Materials availability statement

Materials and reagents newly described in this article can be obtained by contacting the corresponding author.

## Sequencing data

Deposited at Gene Expression Omnibus under accession: GSE243266.

## Cloning

HIV reporter constructs were made by removing the CMV promoter, the entire β-globin sequence, and its PAS from a pcDNA5 FRT/TO plasmid containing the WT β-globin (βWT) gene (*Muniz et al., 2015*) and inserting an HIV promoter and downstream TAR element derived from βΔ5–7 (*Dye and Proudfoot, 2001*). *INTS11* targeting constructs were modified from those we previously described to generate *INTS11-SMASh* cells (*Davidson et al., 2020*). The SMASh tag was removed and replaced with 2xHA dTAG derived from Addgene plasmid 91792 (*Nabet et al., 2018*). Guide RNA expressing Cas9 plasmids to modify *INTS11* or *RBBP6* were made by inserting annealed oligonucleotides, containing the gRNA targeting sequence, into px330 (*Cong et al., 2013*) digested with BbsI. DNA was propagated in *Escherichia coli* DH5α.

## Cell culture and cell lines

HCT116 cells were maintained in DMEM supplemented with penicillin/streptomycin at 37°C, 5% $CO_2$. Cells were regularly checked for mycoplasma contamination. *dTAG-RBBP6* cells were generated using the 'CHoP in' protocol (*Manna et al., 2019*). Briefly, a 24-well dish was transfected with 250 ng of px330 (*Cong et al., 2013*) containing the *RBBP6*-targeting guide and 250 ng of PCR product containing the dTAG degron preceded by a blasticidin or puromycin selection marker (derived from Addgene plasmid 91792 and 91793; *Nabet et al., 2018*). JetPRIME (PolyPlus) was used for transfection. Three days later, cells were passaged into media containing 10 µg/ml blasticidin/1 µg/ml puromycin and colonies were PCR screened ~10 days later. *INTS11-dTAG* cells were generated by homology-directed repair (HDR). A 6-well dish of cells was transfected with 1 µg px330 containing the *INTS11*-targeting guide (described in *Davidson et al., 2020*) and 1 µg each of the HDR repair templates generated as per the above cloning section. Three days later, cells were passaged into media containing 30 µg/ml hygromycin and 800 µg/ml G418. Approximately 10 days later, colonies were picked and then screened by PCR. 1 µM dTAGv-1 (Tocris) was used for 1–14 hr (see figure legends for timings used in each experiment); CDK9i (NVP-2) was used at 250 nM for 2.5 hr; DRB was used at 100 µM for 2.5 hr; THZ2 was used at 5 µM for 2.5 hr.

## POINT-seq and sPOINT-seq

For POINT-seq, we followed the protocol provided in *Sousa-Luís et al., 2021*. The only modification was that we started with a confluent 10 cm dish of cells and performed the immunoprecipitation with 6 µg of anti-RNAPII. Approximately 2% cell volume of *Drosophila* S2 cells was included as a spike in control. Libraries were prepared using the NEBNext Ultra II Directional RNA Library Prep Kit for Illumina (New England Biolabs). sPOINT was performed in the same manner with the following differences. A confluent 15 cm dish of cells and 10 µg of anti-RNAPII were used. In sPOINT, the immunoprecipitated and DNAse-treated RNA was treated with Terminator 5-Phosphate-Dependent Exonuclease (lucigen) to remove any 5' monophosphate-containing transcripts, and libraries were prepared with the SMARTer smRNA-Seq Kit for Illumina (Takara Bio) to selectively capture transcripts <150 nts.

## ChIP-seq

For each experiment, 1 × 10 cm dish of cells was used. Protein:DNA crosslinks were formed by adding formaldehyde (1% v/v) to culture media for 10 min, then quenching with 125 mM glycine. Cells were rinsed 2× with PBS, scraped off the dish, and pelleted in 10 ml PBS at 500 × $g$ for 5 min. We then employed the simple ChIP enzymatic kit (Cell Signalling Technologies) to fragment chromatin and purify RNAPII-bound DNA. We followed the kit protocol except for conjugating 5 µg of anti-total RNAPII to sheep anti-mouse dynabeads (Life Technologies). Sequencing libraries were generated using the NEBNext Ultra II DNA Library Prep Kit for Illumina.

## Total/chromatin-associated RNA isolation and qRT-PCR

For total RNA, a 24-well dish of cells was transfected with 100 ng HIV reporter plasmid and, where co-transfected, 50 ng TAT plasmid (*Adams et al., 1988*) using JetPRIME (PolyPlus) following the manufacturer's protocol. Culture media was refreshed 5 hr post-transfection and dTAGv-1 was added where appropriate. The next day, total RNA was isolated using Trizol (Thermo Fisher) following the manufacturer's protocol. For chromatin-associated RNA, where 4hr dTAGv-1 treatment was employed (for *Figure 4—figure supplement 1F*), a 6-well dish of cells was transfected with 300 ng HIV reporter plasmid and 150 ng TAT. Pelleted cells were resuspended in 800 µl hypotonic lysis buffer (HLB: 10 mM Tris-HCl [pH 7.5], 10 mM NaCl and 2.5 mM $MgCl_2$, 0.5% NP40) and under layered with 200 µl HLB + 10% sucrose. Nuclei were isolated by centrifugation for 5 min at 500 × $g$. These were resuspended in 100 µl NUN1 (20 mM Tris-HCl [pH 7.9], 75 mM NaCl, 0.5 mM EDTA, and 50% glycerol). After addition of 1 ml of NUN2 (20 mM HEPES-KOH [pH 7.6], 300 mM NaCl, 0.2 mM EDTA, 7.5 mM $MgCl_2$, 1% NP-40, 1 M urea), chromatin was isolated by 10 min incubation on ice followed by 10 min centrifugation at 13,000 rpm. RNA was isolated from the chromatin pellet using Trizol following the manufacturers' instructions. For qRT-PCR of total and chromatin-associated RNA, RNA was DNase treated then 1 µg was reverse transcribed using Protoscript II reverse transcriptase (New England Biolabs). qPCR was performed using LUNA SYBR green reagent (New England Biolabs) on a QIAGEN Rotorgene instrument. For all qRT-PCR analysis qantitative analysis used the ΔCT method. Statistical significance was calculated using a Student's $t$-test.

## Co-immunoprecipitation

For each transfection, a semi-confluent 10 cm dish of cells was transfected with 5 µg of plasmid expressing flag tagged wild type or E203Q INTS11. Immunoprecipitation was based on the ELCAP protocol (*Gregersen et al., 2022*). The following day, cells were washed 2× in PBS, scrapped into 10 ml PBS and spun down for 5 min at 500 × $g$. Pelleted cells were resuspended in 800 µl hypotonic lysis buffer low NP40 (HLB: 10 mM Tris-HCl [pH 7.5], 10 mM NaCl and 2.5 mM $MgCl_2$, 0.1% NP40) and under layered with 200 µl HLB low NP40 + 10% sucrose. Nuclei were resuspended in 1 ml Chromatin digestion buffer (20 mM HEPES pH 7.9, 1.5 mM $MgCl_2$, 10% [v/v] glycerol, 150 mM NaCl, 0.1% [v/v] NP-40 and 250 U/ml Benzonase) and incubated for 1 hr at 4°C. Debris was pelleted at 13,000 rpm for 10 min and supernatant was incubated with 20 µl of anti-flag magnetic beads (Sigma) for 2 hr at 4°C. Beads were washed 6× with ice-cold chromatin digestion buffer and samples were eluted by boiling in protein loading buffer (100 mM Tris-Cl [pH 6.8] 4% [w/v] sodium dodecyl sulfate 0.2% [w/v] bromophenol blue 20% [v/v] glycerol) before polyacrylamide separation and western blotting.

## qRT-PCR primers and gRNA target sites

See *Supplementary file 1*.

## Bioinformatics

### POINT-Seq alignment and visualisation

We generated a metagene list of genes with no overlapping regions within 10 kb of any other expressed transcription unit (1316). Adapters were removed from raw reads using Trim Galore! and mapped to GRCh38 using HISAT2 using default parameters. Biological replicates were normalised and merged using SAMtools merge. Split strand metagene plots were produced using *Drosophila* spike-in normalised sense and antisense (scaled to –1) bigwig coverage files separately with further graphical processing performed in R. For heat maps, computematrix (DeepTools) was used to generate score files from the normalised bigwig files using the 10 kb non-overlapping gene list. A log2 ratio

(depletion/control) was applied to identify changes in reads. Plots were redrawn in R; the parameters used for each heat map are detailed in figure legends.

### sPOINT

For sPOINT TSS metaplots showing full-length and 5′ derived coverage, gene lists were determined by selecting principal protein-coding transcript isoforms from gencode v42 human annotation – specifically, those containing both 'appris_principal_1' and 'Ensembl_canonical' labels (15301 in total). The top 20% expressed (based on -dTAG) were used to generate meta profiles (3060 TSSs in total). The 684 genes used to exemplify the difference between POINT and sPOINT coverage are the top 50% of expressed genes (based on POINT-seq in *INTS11-dTAG*) cells, which are separated from neighbouring transcription units by at least 10 kb.

### Antisense PAS analysis

Bed files from *Figure 1C* were edited to map the antisense transcript from each gene by fitting each gene's start and end point to 3 kb upstream of their TSS. Sequences underlying these regions were obtained via BEDTools getFASTA (hg38) and consensus PAS (AATAAA) enumerated in R. PAS frequency per transcript was plotted using ggplot2. Heatmaps for transcripts containing >1 or no PAS motifs were plotted with DeepTools.

### ChIP-seq alignment and plotting

Adapters were removed from raw reads using Trim Galore! and mapped to GRCh38 using HISAT2 with default parameters. Reads were also mapped to Dm6 to identify spike-in signal. Reads with MAPQ score of ≤30 were removed with SAMtools (*Li et al., 2009*). Peaks were called using MACS2 in paired-end mode (*Zhang et al., 2008*). Metaplots were created from S2 scaled, merged replicate -log10 q-value bigwig coverage using DeepTools with further processing performed within R. For *Figure 3—figure supplement 1E*, plots used the same gene list described above for sPOINT promoter analyses (see 'sPOINT' section above).

## Acknowledgements

We thank Hiroshi Kimura for the mouse monoclonal antibody to total RNAPII. Our research was funded by a Wellcome Trust Investigator Award to SW (223106/Z/21/Z). This project used the University of Exeter Sequencing Service, and their equipment was funded by the Wellcome Trust (Multi-User Equipment Grant award number 218247/Z/19/Z).

## Additional information

### Funding

| Funder | Grant reference number | Author |
| --- | --- | --- |
| Wellcome Trust | 10.35802/223106 | Steven West |

The funders had no role in study design, data collection and interpretation, or the decision to submit the work for publication. For the purpose of Open Access, the authors have applied a CC BY public copyright license to any Author Accepted Manuscript version arising from this submission.

### Author contributions

Joshua D Eaton, Conceptualization, Data curation, Formal analysis, Investigation, Methodology, Writing – original draft; Jessica Board, Data curation, Formal analysis, Validation, Investigation, Writing – original draft, Writing – review and editing; Lee Davidson, Data curation, Formal analysis, Supervision, Writing – original draft; Chris Estell, Formal analysis, Investigation, Writing – original draft; Steven West, Conceptualization, Resources, Formal analysis, Supervision, Funding acquisition, Validation, Investigation, Methodology, Writing – original draft, Project administration, Writing – review and editing

## Author ORCIDs
Joshua D Eaton ORCID https://orcid.org/0000-0002-2072-9507
Steven West ORCID http://orcid.org/0000-0002-7622-9050

Reviewer #1 (Public Review): https://doi.org/10.7554/eLife.92764.3.sa1
Reviewer #2 (Public Review): https://doi.org/10.7554/eLife.92764.3.sa2
Reviewer #3 (Public Review): https://doi.org/10.7554/eLife.92764.3.sa3
Author response https://doi.org/10.7554/eLife.92764.3.sa4

## Additional files

### Supplementary files
• Supplementary file 1. Oligonucleotide and other DNA sequences used in this study to engineer cell lines and for qRT-PCR.

• MDAR checklist

### Data availability
Sequencing data have been deposited in GEO under accession code GSE243266.

The following dataset was generated:

| Author(s) | Year | Dataset title | Dataset URL | Database and Identifier |
|---|---|---|---|---|
| Eaton J, Board J, Davidson L, Estell C, West S | 2024 | Human promoter directionality is determined by transcriptional initiation and the opposing activities of INTS11 and CDK9 | https://www.ncbi.nlm.nih.gov/geo/query/acc.cgi?acc=GSE243266 | NCBI Gene Expression Omnibus, GSE243266 |

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
