## [Editor Report · eLife assessment]

The **important** study uses a new experimental method to provide **compelling** evidence on how sense- and anti-sense transcription is differentially regulated. The method described here can generally be used to study the alterations in transcription. This article will be of interest to scientists working in the gene regulation community.

---

## [Referee Report · Reviewer #1 (Public Review)]

Summary:

In this manuscript, Eaton et al. examine the regulation of transcription directionality using a powerful genomic approach (more about the methodology below).

Their data challenge the notion that the polyadenylation signal-reading Cleavage and Polyadenylation (CPA) complex is responsible for controlling promoter directionality by terminating antisense transcription. Namely, depletion of the required CPA factor RBBP6 has little effect on antisense transcription measured by POINT. They find instead that initiation is intrinsically preferential in the sense direction and additionally maintained by the activities of an alternative processing complex called Integrator, together with the kinase CDK9. In the presence of CDK9 activity, depletion of Integrator endoribonuclease INTS11 leads to globally increased transcription in the antisense direction, and minor effects in the sense direction. However, CDK9 inhibition reveals that sense transcription is also sensitive to INS11 depletion. The authors suggest that CDK9 activity is stronger in the sense direction, preventing INTS11-mediated premature termination of sense transcripts.

Strengths:

The combination of acute depletion of the studied factors using degron approaches (important to limit possible secondary effects), together with novel and very sensitive nascent transcriptomics methods POINT and sPOINT is very powerful. The applied spike-in normalization means the analysis is more rigorous than most. Using this methodology allowed the authors to revisit the interesting question of how promoter/transcription directionality is determined.

The data quality appears very good and the fact that both global analysis as well as numerous gene-specific examples are shown makes it convincing.

The manuscript is well written and hence a pleasure to read.

Weaknesses:

The bias in transcriptional initiation directionality remains to be elucidated.

Conclusion/assessment:

This important work substantially advances our understanding of the mechanisms governing the directionality of human promoters. The evidence supporting the claims of the authors is compelling, with a.o. the use of advanced nascent transcriptomics including spike-in normalization controls and acute protein depletion using degron approaches.

In my opinion the authors' conclusions are well supported.

Not only the manuscript but also the data generated will be useful to the wide community of researchers studying transcriptional regulation. Also, the POINT-derived novel sPOINT method described here is very valuable and can positively impact work in the field.

---

## [Referee Report · Reviewer #2 (Public Review)]

Summary:

Eaton and colleagues use targeted protein degradation coupled with nascent transcription mapping to highlight a role for the integrator component INST11 in terminating antisense transcription. They find that upon inhibition of CDK9, INST11 can terminate both antisense and sense transcription - leading to a model whereby INST11 can terminate antisense transcription and the activity of CDK9 protects sense transcription from INST11-mediated termination. They further develop a new method called sPOINT which selectively amplifies nascent 5' capped RNAs and find that transcription initiation is more efficient in the sense direction than in the antisense direction. This is an excellent paper which uses elegant experimental design and innovative technologies to uncover a novel regulatory step in the control of transcriptional directionality.

Strengths:

One of the major strengths of this work is that the authors endogenously tag two of their proteins of interest - RBBP6 and INST11. This tag allows them to rapidly degrade these proteins - increasing the likelihood that any effects they see are primary effects of protein depletion rather than secondary effects. Another strength of this work is that the authors immunoprecipitate RNAPII and sequence extracted full length RNA (POINT-seq) allowing them to map nascent transcription. A technical advance from this work is the development of sPOINT which allows the selective amplification of 5' capped RNAs < 150 nucleotides, allowing the direction of transcription initiation to be resolved.

Weaknesses:

While the authors provide strong evidence that INST11 and CDK9 play important roles in determining promoter directionality, their data suggests that when INST11 is degraded and CDK9 is inhibited there remains a bias in favour of sense transcription (Figure 4B and C). This suggests that there are other unknown factors that promote sense transcription over antisense transcription and future work could look to identify these.

---

## [Referee Report · Reviewer #3 (Public Review)]

Summary:

Using protein degradation approach, Eaton et al show that INST11 can terminate the sense and anti-sense transcription but higher activity of CDK9 in sense direction protects it from INS11-dependent termination. They developed sPOINT-seq that detects nascent 5'-capped RNA. The technique allowed them to reveal robust transcription initiation of sense-RNA as compared to anti-sense.

Strengths:

The strength of paper is acute degradation of proteins, eliminating the off-target effects. Further, the paper uses elegant approaches such as POINT and sPOINT-seq to measure nascent RNA and 5'-capped short RNA. Together, the combination of these three allowed the authors to make clean interpretations of data.

Weaknesses:

While manuscript is well written, the details on panel is not sufficient. The methods can be more elaborate for better understanding. Additional discussion on how authors findings contradict the existing model of anti-sense transcription termination should be added.

in the revised manuscript, authors have added details on panels and elaborated method and other sections for better understanding.

---

## [Author Response]

The following is the authors’ response to the original reviews.

**Public Reviews:**

**Reviewer #1 (Public Review):**
Summary:In this manuscript, Eaton et al. examine the regulation of transcription directionality using a powerful genomic approach (more about the methodology below). Their data challenge the notion that the polyadenylation signal-reading Cleavage and Polyadenylation (CPA) complex is responsible for controlling promoter directionality by terminating antisense transcription. Namely, depletion of the required CPA factor RBBP6 has little effect on antisense transcription measured by POINT. They find instead that initiation is intrinsically preferential in the sense direction and additionally maintained by the activities of an alternative processing complex called Integrator, together with the kinase CDK9. In the presence of CDK9 activity, depletion of Integrator endoribonuclease INTS11 leads to globally increased transcription in the antisense direction, and minor effects in the sense direction. However, CDK9 inhibition reveals that sense transcription is also sensitive to INS11 depletion. The authors suggest that CDK9 activity is stronger in the sense direction, preventing INTS11-mediated premature termination of sense transcrpts.Strengths:The combination of acute depletion of the studied factors using degron approaches (important to limit possible secondary effects), together with novel and very sensitive nascent transcriptomics methods POINT and sPOINT is very powerful. The applied spike-in normalization means the analysis is more rigorous than most. Using this methodology allowed the authors to revisit the interesting question of how promoter/transcription directionality is determined.The data quality appears very good and the fact that both global analysis as well as numerous gene-specific examples are shown makes it convincing.The manuscript is well written and hence a pleasure to read.

We appreciate this positive assessment.

Weaknesses:I am slightly worried about the reproducibility of the data - it is unclear to me from the manuscript if and which experiments were performed in replicate (lack of table with genomic experiments and GEO access, mentioned in more detail in below recommendations to authors), and the methods could be more detailed.

All sequencing data was deposited with GEO. Multiple biological replicates were performed for each sequencing experiment. Bigwig files are presented as a table in the GEO submissions. This data has now been made public.

A separate discussion section would be useful, particularly since the data provided challenge some concepts in the field. How do the authors interpret U1 data from the Dreyfuss lab in light of their results? How about the known PAS-density directionality bias (more PAS present in antisense direction than in sense) - could the differential PAS density be still relevant to transcription directionality?

As suggested, we have expanded our discussion to relate our findings to existing data. We think the results from the Dreyfuss lab are very important and highlight the role of U1 snRNA in enforcing transcriptional elongation. It does this in part by shielding PAS sequences. Recent work from our lab also shows that U1 snRNA opposes the Restrictor complex and PNUTS, which otherwise suppress transcription (Estell et al., Mol Cell 2023). Most recently, the Adelman lab has demonstrated that U1 snRNA generally enhances transcription elongation (Mimoso and Adelman., Mol Cell 2023). Our work does not challenge and is not inconsistent with these studies.

The role of U1 in opposing PAS-dependent termination inspired the idea that antisense transcriptional termination may utilise PASs. This was because such regions are rich in AAUAAA and comparatively poor in U1 binding sites. However, our RBBP6 depletion and POINT-seq data suggest that PAS-dependent termination is uncommon in the antisense direction. As such, other mechanisms suppress antisense transcription and influence promoter directionality. In our paper, we propose a major role for the Integrator complex.

We do not completely rule out antisense PAS activity and discuss the prior work that identified polyadenylated antisense transcripts. Nevertheless, this was detected by oligo-dT primed RT-PCR/Northern blotting, which cannot determine the fraction of non-polyadenylated RNA that could result from PAS-independent termination (e.g. by Integrator). To do that requires an analysis of total nascent transcription as achieved by our POINT-seq. Based on these experiments, Integrator depletion has a greater impact on antisense transcription than RBBP6 depletion.

I find that the provided evidence for promoter directionality to be for the most part due to preferential initiation in the sense direction should be stressed more. This is in my eyes the strongest effect and is somehow brushed under the rug.

We agree that this is an important finding and incorporated it into the title and abstract. As the reviewer recommends, we now highlight it further in the new discussion.

References 12-17 report an effect of Integrator on 5' of protein-coding genes, while data in Figure 2 appears contradictory. Then, experiments in Figure 4 show a global effect of INST11 depletion on promoter-proximal sense transcription. In my opinion, data from the 2.5h time-point of depletion should be shown alongside 1.5h in Figure 2 so that it is clear that the authors found an effect similar to the above references. I find the current presentation somehow misleading.

We are grateful for this suggestion and present new analyses demonstrating that our experiment in Figure 2 concurs with previous findings (Supplemental Figures 2A and B). Our original heatmap (Figure 2E) shows a very strong and general antisense effect of INTS11 loss. On the same scale, the effects in the sense direction are not as apparent, which is also the case using metaplots. New supplemental figure 2A now shows sense transcription from this experiment in isolation and on a lower scale, demonstrating that a subset of genes shows promoter-proximal increases in transcription following INTS11 depletion. This is smaller and less general than the antisense effect but consistent with previous findings. Indeed, our new analysis in supplemental figure 2B shows that affected protein-coding genes are lowly expressed, in line with Hu et al., Mol Cell 2023. This explains why a sense effect is not as apparent by metaplot, for which highly expressed genes contribute the most signal.

As a result of our analyses, we are confident that the apparently larger effect at the 2.5hr timepoint (Figure 4) that we initially reported is due to experimental variability and not greater effects of extended INTS11 depletion. Overlaying the 1.5h and 2.5h datasets (Supplemental Figure 4B) revealed a similar number of affected protein-coding genes with a strong (83%) overlap between the affected genes. To support this, we performed qPCR on four affected protein-coding transcripts which revealed no significant difference in the level of INTS11 effect after 2.5h vs 1.5h (Supplemental Figure 4C).

We now present data for merged replicates in Figures 2 and 4 which reveal very similar average profiles for -INTS11 vs +INTS11 at both timepoints. Overall, we believe that we have resolved this discrepancy by showing that it amounts to experimental variability and because the most acutely affected protein-coding genes are lowly expressed. As detailed above, we show this in multiple ways (and validate by qPCR) We have revised the text accordingly and removed our original speculation that differences reflected the timeframe of INTS11 loss.

Conclusion/assessment:This important work substantially advances our understanding of the mechanisms governing the directionality of human promoters. The evidence supporting the claims of the authors is compelling, with among others the use of advanced nascent transcriptomics including spike-in normalization controls and acute protein depletion using degron approaches.In my opinion, the authors' conclusions are in general well supported.Not only the manuscript but also the data generated will be useful to the wide community of researchers studying transcriptional regulation. Also, the POINT-derived novel sPOINT method described here is very valuable and can positively impact work in the field.

We are grateful for the reviewers' positive assessment of our study.

**Reviewer #2 (Public Review):**
Summary:Eaton and colleagues use targeted protein degradation coupled with nascent transcription mapping to highlight a role for the integrator component INST11 in terminating antisense transcription. They find that upon inhibition of CDK9, INST11 can terminate both antisense and sense transcription - leading to a model whereby INST11 can terminate antisense transcription and the activity of CDK9 protects sense transcription from INST11-mediated termination. They further develop a new method called sPOINT which selectively amplifies nascent 5' capped RNAs and find that transcription initiation is more efficient in the sense direction than in the antisense direction. This is an excellent paper that uses elegant experimental design and innovative technologies to uncover a novel regulatory step in the control of transcriptional directionality.Strengths:One of the major strengths of this work is that the authors endogenously tag two of their proteins of interest - RBBP6 and INST11. This tag allows them to rapidly degrade these proteins - increasing the likelihood that any effects they see are primary effects of protein depletion rather than secondary effects. Another strength of this work is that the authors immunoprecipitate RNAPII and sequence extracted full-length RNA (POINT-seq) allowing them to map nascent transcription. A technical advance from this work is the development of sPOINT which allows the selective amplification of 5' capped RNAs < 150 nucleotides, allowing the direction of transcription initiation to be resolved.

We appreciate this positive assessment.

Weaknesses:While the authors provide strong evidence that INST11 and CDK9 play important roles in determining promoter directionality, their data suggests that when INST11 is degraded and CDK9 is inhibited there remains a bias in favour of sense transcription (Figures 4B and C). This suggests that there are other unknown factors that promote sense transcription over antisense transcription and future work could look to identify these.

We agree that other (so far, unknown) factors promote sense transcription over antisense, which was demonstrated by our short POINT. We have provided an expanded discussion on this in the revision. In our opinion, demonstrating that sense transcription is driven by preferential initiation in that direction is a key finding and we agree that the identification of the underlying mechanism constitutes an interesting avenue for future study.

**Reviewer #3 (Public Review):**
Summary:Using a protein degradation approach, Eaton et al show that INST11 can terminate the sense and anti-sense transcription but higher activity of CDK9 in the sense direction protects it from INS11-dependent termination. They developed sPOINT-seq that detects nascent 5'-capped RNA. The technique allowed them to reveal robust transcription initiation of sense-RNA as compared to anti-sense.Strengths:The strength of the paper is the acute degradation of proteins, eliminating the off-target effects. Further, the paper uses elegant approaches such as POINT and sPOINT-seq to measure nascent RNA and 5'-capped short RNA. Together, the combination of these three allowed the authors to make clean interpretations of data.

We appreciate this positive assessment.

Weaknesses:While the manuscript is well written, the details on the panel are not sufficient. The methods could be elaborated to aid understanding. Additional discussion on how the authors' findings contradict the existing model of anti-sense transcription termination should be added.

We have added more detail to the figure panels, which we hope will help readers to navigate the paper more easily. Specifically, the assay employed for each experiment is indicated in each figure panel. As requested, we provide a new and separate discussion section in the revision.

**Recommendations for the authors:**

**Reviewer #1 (Recommendations For The Authors):**
Congratulations on this important piece of work!Some specific suggestions.MAJOR-The data are not available (Accession "GSE243266" is currently private and is scheduled to be released on Sep 01, 2026.) This should be corrected and as a minimum, the raw sequencing files as well as the spike-in scaled bigwig files should be provided in GEO.

We have made the data public. Raw and bigwig files are provided as part of the GEO upload.

MINOR- It would be useful for readers if you could include catalog numbers of the reagents used in the study.

We have included this information in our revision.

- A table in experimental procedures summarizing the genomic experiments performed in this study as well as published ones reanalyzed here would be helpful.

This is now provided as part of the resources table.

- It would be easier for reviewers to evaluate the manuscript if the figure legends were included together with the figures on one page. This is now allowed by most journals.

We have used this formatting in the revision.

- Providing some captions for the results sections would be helpful.

We have included subheadings as suggested.

**Reviewer #2 (Recommendations For The Authors):**
Generally, I would suggest writing the experiment-type above panels where it is not immediately obvious what they are so a reader can appreciate the figures without referencing the legend. E.g. write POINT-seq on Figure 1B just to make it obvious to someone looking at the figures what methodology they are looking at. Likewise, you could write RNAPII ChIP-seq for Supplementary Figures 3D and 3E.

We have carried out this recommendation.

Can a y-axis be indicated on POINT-seq genome browser tracks? This could make them easier to interpret.

Y-axis scales are provided as RPKM as stated in the figure legends.

The authors could address/speculate in the text why there is less POINT-seq signal for the antisense transcript in the treatment condition in Figure 1B? Or could consider including a different example locus where this is not the case for clarity.

Acute depletion of poly(A) factors (like RBBP6) results in a strong read-through beyond the poly(A) signal of protein-coding genes as Figure 1 shows. However, it also causes a reduction in transcription levels, which can be seen in the figure and is correctly noted by the reviewer in this comment. We see this with other poly(A) factor depletions (e.g. CPSF73 and CPSF30 – Eaton et al., 2020 and Estell et al., 2021) and other labs have observed this too (e.g for CPSF73-dTAG depletion (Cugusi et al., Mol Cell 2022)). Plausible reasons include a limited pool of free RNAPII due to impaired transcriptional termination or limited nucleotide availability due to their incorporation within long read-through transcripts. For these reasons, we have retained the example in Figure 1B as a typical representation of the effect. Moreover, the heatmap in Figure 1D fairly represents the spectrum of effects following RBBP6 loss – highlighting the strong read-through beyond poly(A) signals and the marginal antisense effects.

"The established effect of INTS11 at snRNAs was detected in our POINT-seq data and demonstrates the efficacy of this approach (Figure 2B)." The authors could explain this point more clearly in the text and describe the data - e.g. As expected, depletion of INTS11 leads to increased POINT-seq signal at the 3' end of snRNAs, consistent with defects in transcriptional termination. This is highlighted by the RNU5A-1 and RNU5B-1 loci (Figure 2B).

We agree and have added more context to clarify this.

I would suggest adjusting the scale of the heatmap in Figure 2E - I think it would be easier to interpret if the value of 0 was white - with >0 a gradient of orange and <0 a gradient of blue (as is done in Figure 1C). I think making this change would make the point as written in the text clearer i.e. "heatmap analysis demonstrates the dominant impact of INTS11 on antisense versus sense transcription at most promoters (Figure 2E)." I'm assuming most of the sense transcription would be white (more clearly unchanging) when the scale is adjusted.

We agree and have done this. The reviewer is correct that most sense transcription is unchanged by INTS11 loss. However, as we alluded to in the original submission, a subset of transcripts shows a promoter-proximal increase after INTS11 depletion. We have expanded the analyses of this effect (see responses to other comments) but stress that it is neither as general nor as large as the antisense effect.

The authors make the point that there is mildly increased transcription over the 5' end of some genes upon INST11 depletion and show a track (Supplementary Fig 2A). It is not immediately obvious from the presentation of the meta-analysis in Figure 2D how generalisable this statement is. Perhaps the size of the panel or thickness of the lines in Figure 2D could be adjusted so that the peak of the control (in blue) could be seen. Perhaps an arrow indicating the peak could be added? I'm assuming the peak at the TSS is slightly lower in the control compared to INST11 depletion based on the authors' statement.

We have provided multiple new analyses of this data to highlight where there are promoter-proximal effects of INTS11 loss in the sense direction. Please see our response to the public review of reviewer 1 and new supplemental figures 2A, 2B, 4A and 4B which highlight the sense transcription increased in the absence of INTS11.

The authors label Figure 4 "Promoters lose their directionality when CDK9 is inhibited" - but in INST11 depleted cells treated with CDK9i they find that there still is a bias towards sense transcription. Suggested edit "Some promoter directionality is lost when CDK9 is inhibited" or similar.

We agree and have made this change.

The authors conclude that INTS11-mediated effects are the result of perturbation of the catalytic activities of Integrator, the authors should perform rescue experiments with the catalytically dead E203Q-INTS11 mutant.

This is a very good suggestion and something we had intended to pursue. However, as we will describe below (and shown in Supplemental Figure 4G), there were confounding issues with this experiment.

The E203Q mutant of INTS11 is widely used in the literature to test for catalytic functions of INTS11. However, we have found that this mutation impairs the ability of INTS11 to bind other Integrator modules in cells. Based on co-immunoprecipitation of flag-tagged WT and E203Q derivatives, INTS1 (backbone module), 10 (tail module), and 8 (phosphatase module) all show reduced binding to E203Q vs. WT. Because E203Q INTS11 is defective in forming Integrator complexes, rescue experiments might not fully distinguish the effects of INTS11 activity from those caused by defects in complex assembly. While this may at first seem unexpected, in the analogous 3’ end processing complex, catalytic mutants of CPSF73 (which is highly related to INTS11) negatively affect its interaction with other complex members (Kolev and Steitz, EMBO Reports 2005).

We hypothesise that INTS11 activity is most likely involved in attenuating promoter-proximal transcription, but we cannot formally rule out other explanations and discuss this in our revision. Regardless of how INTS11 attenuates transcription, our main conclusion is on its requirement to terminate antisense transcription whether this involves its cleavage activity or not.

The authors suggest that CDK9 modulates INTS11 activity/assembly and suggest this may be related to SPT5. Is there an effect of CDK9 inhibition on the snRNA's highlighted in Figure 2B?

We believe that snRNAs are different from protein-coding genes concerning CDK9 function. Shona Murphy’s lab previously showed that, unlike protein-coding genes, snRNA transcription is insensitive to CDK9 inhibition, and that snRNA processing is impaired by CDK9 inhibition (Medlin et al., EMBO 2003 and EMBO 2005). We reproduce these findings by metaanalysis of 15 highly expressed and well-separated snRNAs and by qRT-PCR of unprocessed RNU1-1, RNU5A-1 and RNU7-1 snRNA following CDK9 inhibition. We observe snRNA read-through by POINT-seq following INTS11 loss whether CDK9 is inhibited or not (left panel, below). Note the higher TES proximal signal in CDK9i conditions, which likely reflects the accumulation of unprocessed snRNA as validated by qPCR for three example snRNAs (right panel, below).

**Author response image 1. sa4fig1:** 

For Figure 4, would similar results be observed using inhibitors targeting other transcriptional CDKs such as CDK7,12/13?

In response to this suggestion, we analysed four selected protein-coding transcripts (the same 4 that we used to validate the CDK9i results) by qRT-PCR in a background of CDK7 inhibition using the THZ2 compound (new Supplemental Figure 4E). THZ2 suppresses transcription from these genes as expected. Interestingly, expression is restored by co-depleting Integrator, recapitulating our findings with CDK9 inhibition. As CDK7 is the CDK-activating kinase for CDK9, its inhibition will also inhibit CDK9 so THZ2 may simply hit this pathway upstream of where CDK9 inhibitors. Second, CDK7 may independently shield transcription from INTS11. We allude to both interesting possibilities.

What happens to the phosphorylation state of anti-sense engaged RNAPII when INTS11 is acutely depleted and/or CDK9 is inhibited? This could be measured by including Ser5 and Ser2 antibodies in the sPOINT-seq assay and complemented with Western Blot analysis.

We have performed the western blot for Ser5 and Ser2 phosphorylation as suggested. Both signals are mildly enhanced by INTS11 loss, which is consistent with generally increased transcription. Ser2p is strongly reduced by CDK9 inhibition, which is consistent with the loss of nascent transcription in this condition. Interestingly, both modifications are partly recovered when INTS11 is depleted in conjunction with CDK9 inhibition. This is consistent with the effects that we see on POINT-seq and shows that the recovered transcription is associated with some phosphorylation of RNAPII CTD. This presumably reflects the action(s) of kinases that can act redundantly with CDK9.

We have not performed POINT-seq with Ser5p and Ser2p antibodies under these various conditions. Our rationale is that our existing data uses an antibody that captures all RNAPII (regardless of its phosphorylation status), which we feel most comprehensively assays transcription in either direction. Moreover, the lab of Fei Chen (Hu et al., Mol Cell 2023) recently published Ser5p and Ser2p ChIP-seq following INTS11 loss. By ChIP-seq, they observe a bigger increase in antisense RNAPII occupancy vs. sense providing independent and orthogonal support for our POINT-seq data. Interestingly, this antisense increase is not paralleled by proportional increases in Ser5p or Ser2p signals. This suggests that the unattenuated antisense transcription resulting from INTS11 loss does not have high Ser5p or Ser2p. Since CDK7 and 9 are major Ser5 and 2 kinases, this supports our model that their activity is less prevalent for antisense transcription. We now discuss these data in our revision.

The HIV reporter RNA experiments should be performed with the CDK9 inhibitor added to the experimental conditions. Presumably CDK9 inhibition would result in no upregulation of the reporter upon addition of TAT and/or dTAG. Perhaps the amount of TAT should be reduced to still have a dynamic window in which changes can be detected. It is possible that reporter activation is simply at a maximum. Can anti-sense transcription be measured from the reporter?

We have performed the requested CDK9 inhibitor experiment to confirm that TAT-activated transcription from the HIV promoter is CDK9-dependent (new supplemental figure 4F). Consistent with previous literature on HIV transcription, CDK9 inhibition attenuates TAT-activated transcription. Importantly, and in line with our other experiments, depletion of INTS11 results in significant restoration of transcription from the HIV promoter when CDK9 is inhibited. Thus, TAT-activated transcription is CDK9-dependent and, as for endogenous genes, CDK9 prevents attenuation by INTS11.

While TAT-activated transcription is high, we do not think that the plasmid is saturated. When considering this question, we revisited previous experiments using this system to study RNA processing (Dye et al., Mol Cell 1999, Cell 2001, Mol Cell 2006). In these cases, mutations in splice sites or polyadenylation sites have a strong effect on RNA processing and transcription around HIV reporter plasmids. Effects on transcription and RNA processing are; therefore, apparent in the appropriate context. In contrast, we find that the complete elimination of INTS11 has no impact on RNA output from the HIV reporter. Our original experiment assessing the impact of INTS11 loss in +TAT conditions used total RNA. One possibility is that this allows non-nascent RNA to accumulate which might confound our interpretation of INTS11 effects on ongoing transcription. However, the new experiment described in the paragraph above was performed on chromatin-associated (nascent) RNA to rule this out. This again shows no impact of INTS11 loss on HIV promoter-derived transcription in the presence of TAT.

To our knowledge, antisense transcription is not routinely assayed from plasmids. They generally employ very strong promoters (e.g. CMV, HIV) to drive sense transcription. Crucially, their circular nature means that RNAPII going around the plasmid could interfere with antisense transcription coming the other way which does not happen in a linear genomic context. This is why we restricted our use of plasmids to looking at the effects of stimulated CDK9 recruitment (via TAT) on transcription rather than promoter directionality.

The authors should clearly state how many replicates were performed for the genomics experiments. Ideally, a signal should be quantified and compared statistically rather than relying on average profiles only.

We have stated the replicate numbers for sequencing experiments in the relevant figure legends. All sequencing experiments were performed in at least two biological replicates, but often three. In addition, we validated their key conclusions by qPCR or with orthogonal sequencing approaches.

**Reviewer #3 (Recommendations For The Authors):**
The authors provide strong evidence in support of their claims.ChIP-seq of pol2S5 and S2 upon INST11 and CDK9 inhibition will strengthen the observation that transcription in the sense direction is more efficient.

We view the analysis of total RNAPII as the most unbiased way of establishing how much RNAPII is going one way or the other. Importantly, ChIP-seq was very recently performed for Ser2p and Ser5p RNAPII derivatives in the lab of Fei Chen (Hu et al., Mol Cell 2023). Their data shows that loss of INTS11 increases the occupancy of total RNAPII in the antisense direction more than in the sense direction, which is consistent with our finding. Interestingly, the increased antisense RNAPII was not paralleled with an increase in Ser2p or Ser5p. This suggests that, following INTS11 loss, the unattenuated antisense transcription is not associated with full/normal Ser2p or Ser5p. These modifications are normally established by CDK7 and 9; therefore, this published ChIP-seq suggests that they are not fully active on antisense transcription when INTS11 is lost. This supports our overall model that CDK9 (and potentially CDK7 as suggested for a small number of genes in new Supplemental Figure 4E) is more active in the sense direction to prevent INTS11-dependent attenuation. We now discuss these data in our revision.

In Supplementary Figure 2, the eRNA expression increases upon INST11 degradation, I wonder if the effects of this will be appreciated on cognate promoters? Can the authors test some enhancer:promoter pairs?

We noticed that some genes (e.g. MYC) that are regulated by enhancers show reduced transcription in the absence of INTS11. Whilst this could suggest a correlation, the transcription of other genes (e.g. ACTB and GAPDH) is also reduced by INTS11 loss although they are not regulated by enhancers. A detailed and extensive analysis would be required to establish any link between INTS11-regulated enhancer transcription and the transcription of genes from their cognate promoters. We agree that this would be interesting, but it seems beyond the scope of our short report on promoter directionality.

Line 111, meta plot was done of 1316 genes. Details on this number should be provided. Overall, the details of methods and analysis need improvement. The layout of panels and labelling on graphs can be improved.

We have now explained the 1316 gene set. In essence, these are the genes separated from an expressed neighbour by at least 10kb. This distance was selected because depletion of RBBP6 induces extensive read-through transcription beyond the polyadenylation site of protein-coding genes. To avoid including genes affected by transcriptional read-through from nearby transcription units we selected those with a 10kb gap between them. This was the only selection criteria so is unlikely to induce any unintended biases. Finally, we have added more information to the figure panels and their legends, which we hope will make our manuscript more accessible.